# Analysis of Endangered Andalusian Black Cattle (Negra Andaluza) Reveals Genetic Reservoir for Bovine Black Trunk

**DOI:** 10.3390/ani14071131

**Published:** 2024-04-08

**Authors:** Luis Favian Cartuche Macas, María Esperanza Camacho Vallejo, Antonio González Ariza, José Manuel León Jurado, Juan Vicente Delgado Bermejo, Carmen Marín Navas, Francisco Javier Navas González

**Affiliations:** 1Faculty of Biology, Amazonian State University, Lago Agrio Headquarters, Nueva Loja 210203, Ecuador; luis.cartuche@uaw.edu.ec; 2Department of Genetics, Faculty of Veterinary Sciences, University of Córdoba, 14014 Córdoba, Spain; juanviagr218@gmail.com (J.V.D.B.); carmen95_mn@hotmail.com (C.M.N.); 3Institute of Agricultural Research and Training (IFAPA), Alameda del Obispo, 14005 Córdoba, Spain; mariae.camacho@juntadeandalucia.es; 4Centro Agropecuario Provincial de Córdoba, Diputación Provincial de Córdoba, 14014 Córdoba, Spain; angoarvet@outlook.es (A.G.A.); jomalejur@yahoo.es (J.M.L.J.)

**Keywords:** Andalusian Black cattle breed, conservation, demographics, studbook, inbreeding, Genetic Conservation Index

## Abstract

**Simple Summary:**

This study reveals a significant decline in the Black Andalusian cattle breed population due to agricultural changes and foreign breed introductions, leading to smaller herds and a shift in the male-to-female ratio. Although inbreeding rates are low, there is concern over assortative mating. Key ancestors influence genetic diversity, with variations in population size and geographic subpopulations. Historical transhumance routes, notably the Cañadas Reales, impact genetic connections. Conservation efforts, including breed association initiatives, have increased registrations, but challenges persist, requiring genealogical registration, targeted breeding, and collaborative efforts. Monitoring and adjusting selection practices are crucial to maintaining genetic diversity. Continued proactive measures are essential for conserving the Black Andalusian breed’s genetic viability, acknowledging historical factors, and addressing contemporary challenges.

**Abstract:**

This comprehensive study on the Andalusian Black cattle breed reveals a substantial population decline, with the average herd size decreasing significantly from 305.54 to 88.28 animals per herd. This decline is primarily attributed to agricultural changes and the introduction of foreign meat-focused breeds. The male-to-female ratio shift is noteworthy, with more cows than bulls, impacting selection intensity for both genders. Inbreeding levels, though relatively low historically (5.94%) and currently (7.23%), raise concerns as 37.08% historically and 48.82% currently of the animals exhibit inbreeding. Positive assortative mating is evident, reflected by the increasing non-random mating coefficient (α). Key ancestors play a crucial role in shaping genetic diversity, with one ancestor significantly influencing the current genetic pool and the top 10 ancestors contributing substantially. Breed maintains a conservation index of 2.75, indicating relatively high genetic diversity. Recent conservation efforts have led to an increase in registered animals. The Cañadas Reales, historical transhumance routes, may have contributed to genetic connections among provinces. Challenges include the historical bottleneck, demographic changes, and potential impacts from reproductive practices. The Andalusian Black breed’s conservation necessitates ongoing efforts in genealogical registration, targeted breeding programs, and collaborative initiatives to address the observed demographic shifts and ensure sustainable genetic diversity.

## 1. Introduction

The Andalusian Black cattle breed is located in the Autonomous Community of Andalusia, Spain, from which it takes its name. It is scattered in two main regions: the first in the Sierra Morena areas of Córdoba and Huelva, and the second in the lowlands of Sevilla, Cádiz, and Huelva [1] (Figure 1). Currently, it is classified as an endangered native breed for conservation purposes, with an officially declared population of 1797 animals in 22 officially declared farms as of the year 2020 and an average annual population growth of 6.31% in the period from 2009 to 2020 [2].

Historically, this breed was specialized for agricultural work, but with the advent of agricultural mechanization, its focus shifted towards meat production, leading it to more mountainous areas and making use of mountain pastures. It has contributed to ecological and social balance along with other breeds. As a result, it has great potential for organic meat production, thus improving the income of producers by offering a 100% native quality brand [3].

Animals of this breed are characterized by a coat, skin, and hair of uniform black color with variations of “morcillo” black on the lower legs, testicular region, and udders. They also have black mucous membranes and white horns with black tips (Figure 2 and Figure 3). Their profile is straight with considerable variation towards convexity, moderate proportion, and a tendency towards hypermetry with marked sexual dimorphism (Figure 2 and Figure 3). The established morphological rating system is based on visual evaluation and a point-based method conducted at 24 months of age, with a rating by region from 1 to 10. Animals with a rating below 5 are disqualified. Subsequently, these values are multiplied by weighting coefficients, resulting in classifications from insufficient to excellent [3,4].

Since 2004, with the establishment of the Association of Breeders of the Andalusian Black Cattle Breed and the corresponding approval of conservation programs for the breed in 2012 and 2020 by the Ministry of Agriculture, Fisheries, and Food (MAPA), the breed has made a significant recovery thanks to the implementation of a pedigree book and performance monitoring. The breed conservation program focuses on meat production in extensive systems, with seasonal supplementation for industrial crossbreeding with specialized breeds. As a result, most farms obtain purebred or crossbred calves that are sent to commercial feedlots [3,6].

One of the advantages of this breed is its reproductive aptitude, as it exhibits high fertility and longevity under challenging production conditions, where adaptability and hardiness play a significant role. Females reach their first heat cycle between 18 and 20 months of age, depending on their development and nutrition, achieving an 80% fertility rate and 95% conception rate with a 1.15% incidence of twin births [2,5]. Under traditional weaning, the 6–7-month-old calves weigh between 190 and 220 kg, reaching an average adult weight of 550–600 kg in females and 850–900 kg in males [2].

Established in 1273 by Alfonso X the Wise, the Mesta Council brought together shepherds from León and Castile, forming an Iberian association. Alfonso X granted them significant privileges, including exemption from military service and the right to testify in trials, as well as rites of passage and pasturage. Over time, additional royal privileges were bestowed upon the Mesta, accompanied by protective measures against farmers. This sparked numerous legal disputes throughout the Middle Ages. In 1836, the Mesta was abolished, partly due to conflicts with Portugal, which rendered many livestock routes impassable [7]. 

However, the Cañadas Reales played their part as diversity routes of domestic animal genetic resources from the north of the Iberian peninsula to the south, and even transboundarily into Portugal (Figure 4). These routes played a pivotal role in shaping Iberian livestock breeds. These ancient transhumance routes served as corridors for the seasonal migration of large herds of sheep and parallelly other livestock which joined them for labor or transport, connecting disparate landscapes and fostering genetic exchange among local populations (Figure 5). 

As a result, the movement of livestock along the Cañadas Reales not only influenced the genetic composition of various breeds but also contributed to the establishment of connections and similarities among them. The intricate web of crossbreeding and intermingling that occurred along these routes has left a lasting impact on the diversity and adaptability of Spanish livestock. Today, the legacy of the Cañadas Reales persists in the distinct characteristics and shared traits observed in different breeds, highlighting the enduring influence of historical transhumance on the genetic landscape of Spain’s livestock. 

The transhumance routes not only facilitated the movement of large sheep flocks but also served as pathways for the migration of cattle herds [8]. Cattle played a multifaceted role in medieval Spain, serving not only as a source of meat and dairy but also as essential draft animals for agricultural labor. The intricate network of the Cañadas Reales allowed for the intermingling of cattle breeds from diverse regions, leading to the exchange of genetic material and the emergence of drifted populations. The adaptability and strength of these draft animals were crucial for the agricultural productivity of various regions along the migration routes. As a result, the genetic diversity routes established by the Mesta influenced not only the characteristics of sheep breeds but also shaped the traits and capabilities of cattle, leaving a lasting imprint on the historical and contemporary livestock populations in Spain [9].

The implemented conservation program’s primary objective is to maintain the breed and improve the maternal line. This means preserving genetic diversity levels (minimizing inbreeding effects) and enhancing maternal reproductive efficiency, with a focus on the calving interval. Also, at the same time, it means preserving ancestral reservoirs of genes and their interconnection routes, which are the only resources that can shed light about the process of configuration that endangered breeds were exposed to.

The primary objective of this study is to develop a model for analyzing pedigree completeness in a downward manner, with a focus on understanding the repercussions of ancestors and founders. The aim is to not only assess the structure of the population and its genetic variability but also to uncover the historical routes and directions that genetic diversity followed during the configuration of the breed. By evaluating the connections between genetic diversity and demographic parameters, the model seeks to measure the existing gene flux and quantify the risk of genetic diversity loss. Additionally, this study aims to determine the endangerment degree of authochtonous bovines and other endangered animal small populations. This comprehensive analysis will serve as a foundation for suggesting effective conservation strategies, taking into account the historical context and dynamics that have shaped the genetic landscape of these populations.

## 2. Materials and Methods

### 2.1. Pedigree Database and Software Tool

The pedigree data for the Andalusian Black cattle breed in the Spanish herdbook were provided by the Association of Breeders of the Andalusian Black Cattle Breed. Appendix A offers additional official details regarding the territorial distribution, husbandry practices, reproductive management, and conservation strategies currently in place, along with other pertinent information concerning the Andalusian Black cattle breed.

Three pedigree datasets were used. First, the complete pedigree dataset (historical population) which comprised 8555 dead and alive animals: 3593 bulls and 4962 cows, born between January 1994 and May 2020. Second, the current pedigree dataset (current population) comprising all alive animals in the population of the breed (2472 animals: 593 bulls and 1879 cows, born between February 2003 and May 2020). 

Subsequently, calculations related to genetic diversity, probabilities of gene origin, and founder analyses can only be conducted either by solely considering animals with both parents known or by comparing them to the historical and current datasets, as suggested by Casanovas Arias et al. [10] and Navas et al. [11]. As a result, a third dataset was considered (reference population).

The reference population dataset included 7351 animals from the current population, consisting of 3435 bulls and 3916 cows, with all sires and dams known (first generation completely known). Considering population sets with known first-generation animals may help identify distortions in diversity parameters commonly observed in endangered breeds with incomplete genealogies. Additional details on the sample composition are available in Table 1.

ENDOG (v4.8) software [12] was utilized for conducting demographic and genetic analyses, enabling the quantification and tracing of pedigree diversity back to ancestors. In our study, pedigree errors were meticulously addressed using the (v4.8) software [12]. It is imperative to ensure the coherence of the pedigree file, as the software requires it for proper execution. Any inconsistencies identified were meticulously corrected manually, as each case may necessitate specific correction methods. Errors observed predominantly encompassed sex misattributions, filiation errors, age discrepancies, and animal naming issues.

### 2.2. Population Summary Statistics

The number of births was analyzed to establish both the maximum and average number of offspring per bull or cow. Navas et al.’s [11] criterion for the determination of the Pedigree Completeness Index (PCI) for each dataset was followed. For the four gametic pathways—sire to son, sire to daughter, dam to son, and dam to daughter—the average age of parents at the birth of offspring and generation intervals were computed. This analysis involved utilizing birth date records for each animal and their respective parents. Table 1 showcases a summary of statistics obtained from the pedigree analyses.

The total and mated cow-to-bull ratios were also calculated. These ratios involved dividing the total number of cows by the total number of bulls to obtain the total ratio and dividing the number of breeding cows by the number of breeding bulls to obtain the mated ratio. Furthermore, a summary of this analysis, including details such as the maximum number of traced generations, pedigree completeness (for first, second, third, fourth, and fifth generations of ancestors), number of maximum generations, number of complete generations, and number of equivalent generations in the historical and current datasets, is presented in Table 2 and Appendix A. Details regarding generation intervals (years) and the mean age (years) of the parents at the birth of their offspring selected for breeding for the four gametic routes in the Andalusian Black cattle breed are showcased in Appendix A.

### 2.3. Inbreeding, Coancestry, and Assortative Mating Degree

The coefficient of coancestry (C), or kinship [13], measures how likely it is that genes that are randomly chosen from two individuals come from a shared ancestor. This means that it shows the probability of shared ancestry between two individuals. Inbreeding (F), however, refers to the probability that both copies of a gene in an individual’s offspring come from the same ancestor. This is often termed the coefficient of inbreeding and represents the extent of genetic relatedness within a population.

Consequently, the coefficient of relationship (C) between two individuals represents the coefficient of inbreeding (F) of their potential offspring. Individual F was computed using the methods outlined in Meuwissen and Luo [14]. Each individual’s average coefficient of relatedness (∆R) indicates the likelihood that two related individuals have inherited a specific allele from their common ancestor (referred to as Identical by Descent, IBD), and was calculated following the approach described by Gutiérrez et al. [12]. According to Leroy et al. [15], although F and C are both estimators of IBD, the values for these probabilities may differ depending on whether the alleles considered belong to a single individual or two individuals, respectively. The individual rate of inbreeding ($\bar{∆F}$) for each generation was calculated according to Gutiérrez et al. [16] through ${∆F}_{b}=1-\sqrt[{t_{b}-1}]{1-F_{b}}$, where *t_b_* is the number of complete equivalent generations and *F_b_* is the F of individual *b*. 

The average inbreeding coefficient (F) per generation was utilized to evaluate the explanatory and predictive capabilities of both linear and quadratic regression functions, projecting F 15 generations into the future.

The individual rate of coancestry ($\bar{∆C}$) for each generation was computed following the methods described in Cervantes et al. [17] through $C_{ba}=1-\sqrt[{\frac{tb+ta}{2}}]{1-C_{ba}}$, where *t_b_* and *t_a_* represent the number of equivalent complete generations and *C_ba_* denotes the coancestry for individuals *b* and *a*. Homogamy, assortative or non-random mating degree (α), delineates the mating pattern wherein individuals with similar phenotypes or genotypes preferentially mate with one another over what would be anticipated under a heterogamy, random, or disassortative mating pattern (where individuals with different genotypes or phenotypes are more likely to mate randomly). In animals, assortative mating is less common than disassortative mating. Non-random mating was calculated following the methodology outlined in Caballero and Toro [18], expressed as $\left(1-F\right)=(1-C)(1-\alpha )$. 

### 2.4. Ancestral Contributions and Probabilities of Gene Origin

The effective number of founders (*fe*) was computed using the formula $f_{e}=\frac{1}{\sum_{k=1}^{f}q_{k}^{2}}$, where *q_k_* represents the probability of gene origin of the *k*th founder and *f* is the actual number of founders [19]. The effective number of ancestors (*fa*) was determined by the formula $f_{a}=\frac{1}{\sum_{k=1}^{f}p_{k}^{2}}$, where *pk* denotes the marginal contribution of the *k*th ancestor [20]. The effective number of founder genomes (*fg*) was calculated as the inverse of twice the average coancestry, as per Caballero and Toro [18]. The expected marginal contribution of each major ancestor *j* was computed according to Boichard et al. [20], while the contributions to inbreeding from nodal common ancestors (inbreeding loops) were computed as outlined in Colleau and Sargolzaei [21]. CFC version 1.0 software was utilized to compute ancestral contributions and probabilities of gene origin [22]. 

The mean effective population size ($\bar{N_{e}}$ [23] was derived as $\bar{N_{e}}=\frac{1}{2\bar{∆IBD}}$. The equivalent subpopulation number was calculated following the premises in Cervantes et al. [24] through $S=\frac{\bar{N_{e}C_{i}}}{N_{e}F_{i}}$, where $\bar{N_{e}C_{i}}=\frac{1}{\left(2\bar{∆C}\right)}$ represents the average effective population size considering *C* and $\bar{N_{e}F_{i}}=\frac{1}{\left(2\bar{∆F}\right)}$ is the average effective population size calculated considering an individual increase in the *F* rate. Genetic diversity (*GD*) was calculated using $GD=1-\frac{1}{2f_{g}}$. *GD* lost in the population since the founder generation was estimated using 1−GD. The *GD* loss attributed to the unequal contribution of founders was estimated following Caballero and Toro [18] using $1-{GD}^{*}$, where ${GD}^{*}=1-\frac{1}{2f_{e}}$. The disparity between *GD* and *GD** indicates the *GD* loss attributed to genetic drift accumulated since the population’s foundation [19], and the effective number of non-founders (*N_ef_*) was calculated using $N_{ef}=\frac{1}{\frac{1}{f_{ge}}-\frac{1}{f_{e}}}$ employing the formula outlined by Caballero and Toro [18]. 

### 2.5. Herd, Municipality, and Province Relationships 

The relationships between herds, municipalities, and provinces were assessed through Nei’s minimum genetic distance [25] between subpopulations i and j, computed as described in Navas et al. [11] to evaluate inter-herd, inter-municipal, and inter-provincial relationships. Wright’s F-statistics, or fixation indexes, which describe the statistically expected degree of heterozygosity reduction when compared to Hardy–Weinberg Equilibrium (HWE) expectations, were computed following the principles outlined in Caballero and Toro [18].

The F-statistics include F_IS_ (or F relative to the subpopulation), F_ST_ (the correlation between random gametes drawn from the subpopulation relative to the total population), and F_IT_ (F relative to the total population). F_IT_ can be decomposed into F_ST_, attributed to the Wahlund effect (a reduction in heterozygosity due to subpopulation structure), and F_IS_, resulting from inbreeding.

Furthermore, self-coancestry (f) was calculated as the probability that two alleles randomly selected from an individual (independently and with replacement) are Identical by Descent (IBD).

Considering that the Identical by Descent (IBD) condition can arise from either sampling the same allele twice or sampling two alleles that happen to be IBD, the coancestry of an individual with itself, denoted as f(A/A), is calculated as $\frac{1+F(A)}{2}$, where *F*(*A*) represents the inbreeding coefficient for that individual. It is crucial to recognize that coancestry in one generation translates into inbreeding in the subsequent one.

Subsequently, selfing, or autocoancestry, should be addressed. Selfing occurs when an individual mates with itself, a scenario often associated with hermaphroditism. In such cases, selfing equals the inbreeding coefficient (*F*) when autofecundation occurs. However, in the context of animal populations, the possibility of autofecundation is typically dismissed and considered to be 0, as is the case in our analysis.

## 3. Results

### 3.1. Population Summary Statistics

The number of herds (28) has remained stable since the registration of the first animal in the pedigree record. Table 1 provides a summary of statistics obtained from pedigree analysis in both the historical (n = 8555) and current (n = 2472) populations. Additionally, the average herd size has decreased from 305.54 to 88.28 animals per herd in the historical and current population datasets, respectively.

While, historically, the total percentage of bulls and cows constituted 42.00% and 58.00%, respectively, these percentages have shifted to 23.99% for bulls and 76.01% for cows in the current population. Appendix A presents descriptive statistics for the average age (years) of parents at the birth of their offspring and generation intervals (years) for the four gametic routes in the Andalusian Black cattle breed. The average generation interval for the historical population was 5.97 years, while it slightly increased to 6.04 years for the current population.

Moreover, the average number of calves per mated bull increased from 2.05 to 3.87 and the average number of calves per mated cow increased from 1.53 to 1.64, from the historical to the current population, respectively. The ratio of cows-to-bulls mated increased from 1.38/1 to 3.17/1.

Furthermore, male selection intensity decreased from 11.37% to 0.58% and female selection intensity decreased from 40.57% to 32.32%, from the historical to the current population. Maximum progeny per bull (379) and cow (14) in the historical population moderately decreased in the current population to 281 per mated bull and slightly decreased to 13 per mated cow, respectively, as detailed in Table 1.

### 3.2. Inbreeding, Coancestry, and Degree of Non-Random Mating

The average inbreeding coefficient (*F*) was relatively low, standing at 5.94% in the historical population and 7.23% in the current population. Despite the presence of a very small proportion of highly inbred animals (maximum F of 43.75%), the percentages of inbred animals were notably high, reaching 37.08% and 48.82% in the historical and current populations, respectively, although they did not reach alarming levels of inbreeding. The average coancestry coefficient (*C*) was 2.90% and 3.06%, and the degree of non-random mating (*α*) exhibited a progressively increasing trend, with values of 0.049 and 0.061 for the historical and current population sets, respectively (see Table 3). The average inbreeding coefficient peaked at 9.48% in 2016, before declining to 7.53% in 2020, while the maximum average coancestry of 1.41% was observed in 2016.

The milestones occurring along the process of configuration, development, and consolidation of the breed can be found in Appendix A. The average degree of non-random mating reached a maximum of 0.083 in 2017. Matings between highly inbred animals, 24 (0.28%) matings between full sibs, 1322 (15.45%) matings between half sibs, and 775 (9.06%) parent–offspring matings have occurred. Table 3 presents the results for average *F*, ΔF, maximum *F*, inbred and highly inbred animals (%), *C*, ΔR, assortative or non-random mating rate (α), and GCI.

Figure 6 represents the evolution of the average coefficient of inbreeding (F, %), average coancestry (C, %), average individual increase in inbreeding (ΔF, %), average relatedness (∆R, %), and degree of non-random mating (α) of the Andalusian Black cattle breed population according to the number of maximum, complete and equivalent generations from 1994 to 2020.

### 3.3. Probabilities of Gene Origin and Ancestral Contributions

Table 4 presents a summary of measures concerning genetic variability and an analysis of gene origin, including the effective number of non-founders (*Nef*), number of founder equivalents (*fe*), and effective number of ancestors (*fa*).

The results for the analysis of genetic diversity loss are presented in Table 5. 

In terms of marginal genetic contribution, a sole ancestor with the identification number 593 was found to account for up to 7.73% of the genetic makeup within the reference population, consisting of animals with known parentage. Within the current population, this ancestor contributed to approximately 10.56% of the genetic diversity. Additionally, this ancestor marginally contributed to 0.45% of total inbreeding and 0.10% of total coancestry, predominantly through the formation of inbreeding loops via nodal common ancestors.

The top 10 ancestors collectively contributed to 3.23% of total inbreeding and 0.68% of total coancestry. Remarkably, they explained a substantial portion, 41.71%, of the overall genetic diversity.

The effective population size, calculated based on individual inbreeding rates (NeFi), was determined to be 12.6101 ± 3.2523. Conversely, when computed based on individual coancestry rates (NeCi), the effective population size was estimated at 45.9251 ± 3.0945. This computation involved analyzing 1,076,400 coancestries between animals of different sexes. Furthermore, the number of equivalent subpopulations was identified to be 3.6419 ± 0.9700.

### 3.4. Herd, Municipality, and Province Relationships

The 28, 23, and 5 existing subpopulations were computed considering herds, municipalities, and provinces as the subdivision criteria. The mean number of animals per herd was 305.54, per municipality was 371.96, and per province was 1711.00. A total of 378, 253, and 10 Nei’s genetic distances were computed considering herds, municipalities, and provinces (Table 6). Nei’s average genetic distance between them was 0.084, 0.068, and 0.011 for herds, municipalities, and provinces, respectively. Mean coancestry within subpopulations was 9.50%, 7.90%, and 2.20% and mean inbreeding was 9.50%, 7.80%, and 2.20%, while the mean coancestry in the metapopulation was 1.20%, 1.10%, and 1.1% and the self-coancestry was 53.1%, 53.0%, and 53.0% for herds, municipalities, and provinces, respectively.

Studying Wright F parameters, the *F* values relative to the total population (*F*_IT_) were 0.052, 0.049, and 0.049 when herds, municipalities, and provinces were considered. The *F* values relative to the subpopulation (*F*_IS_) were 0.036, 0.021, and 0.039 when herds, municipalities, and provinces were considered. The correlations between random gametes drawn from the subpopulation relative to the total population (*F*_ST_) were 0.085, 0.069, and 0.011 when herds, municipalities, and provinces were considered (Table 6). The assessment of the herds, municipalities, and provinces structures revealed that none of them could be considered the population nucleus. The number of herds, municipalities, and provinces which did not use own-bred sires was almost one-third times lower than the number of those that did, and none of them (herds, municipalities, and provinces) were totally isolated. Jaén and Cádiz and Carcabuey and Cabra were the provinces and municipalities more genetically distant (0.1130 and 0.2226). Huelva and Córdoba were the genetically closest provinces (Nei’s minimum distance/average homozygosity of 0.0147), while Alajar and Villanueva de Córdoba (located in the same respective provinces) were the genetically closest ones (Nei’s minimum distance/average homozygosity of 0.0283). One pair of herds/breeders held the greatest Nei’s genetic distance between them (0.1907), while the shortest distance was 0.0192 between one pair of herds/breeders (Nei’s minimum distance/average homozygosity). A dendrogram displaying the relationship between flocks through Nei’s genetic distances is shown in Figure 7 and Figure 8, where all the relationships among herds/breeders, municipalities, and provinces are represented. 

## 4. Discussion

The Andalusian Black breed experienced a significant population decline historically due to agricultural mechanization and the introduction of foreign meat-focused breeds. Initially described between 1918 and 1944, it was officially recognized as a breed between 1981 and 2002 in the Spanish Livestock Breeds Catalog. Its original focus on both work and meat was altered by these changes [2,21].

The Association of Breeders of the Andalusian Black Cattle Breed was founded in 2004 and has managed the pedigree book since 2005. Since 2012, it has been classified as an endangered native breed, prompting a conservation program. As a result, registered animals have risen from 1252 in 2009 to 2121 by 2020 [2]. Presently, due to its reproductive characteristics, the breed is used for breeding pure animals and for crossbreeding with specialized breeds for veal production [26].

The Andalusian Black cattle breed has remained in existence across five provinces (Seville, Córdoba, Huelva, Jaén, and Cádiz), spanning 22 municipalities and 28 herds. In contrast, the Mertolenga breed in Portugal is present in 700 herds, with 216 under genealogical control [27]; the Alentejanean breed is distributed into 120 herds [28]; and the Maremmana breed, in Italy, is distributed among 309 herds [29], highlighting a difference in conservation efforts.

The average number of animals per herd in the Andalusian Black breed decreased significantly from 305.54 in the historical population to 88.28 in the current population (2003–2020). In contrast, the Mertolenga breed has an average of 103.5 registered cows per herd, showing a difference in population size trends [27]. On the other hand, with the Alentejana breed, a similar value of 85.5 ± 63.4 registered cows is observed [28]. The main cause of this decrease is due to the introduction of foreign breeds (replacement) as well as the use of females for crossbreeding (terminal fattening calves), as has occurred in the Polish Red breed [30,31].

In the current population of the Andalusian Black breed, there has been a decrease in the number of bulls and an increase in the number of cows, reflected in the cow-to-bull ratio. This ratio has shifted from 1.38 to 3.17, while the cow-to-bull paired ratio has risen from 26.1 to 36.71. This contrasts with the Polish Red breed, which has a reported ratio of 9.3 [32] and is a breed of international significance because it is the only native breed in this country with the characteristics of resistance and adaptability, such as a high content of total solids, resistance to harsh environments and diseases (tuberculosis and mastitis), good fertility, and longevity [30].

Likewise, the male-to-female ratio’s impact on the short-term survival of founder genes is noteworthy, although population contraction or expansion and family size variation have more substantial effects [33]. The intensity of selection may contribute to the reduction of this ratio, as seen in the Indubrasil and Gyr breeds from 1993 to 2001 in Brazil [34,35].

Regarding parameters such as the average number of offspring per paired bull, it increased by 11.58 (from 80.18 to 91.76), with a reduction in cows of 2.65 (from 11.24 to 8.59). These figures surpassed those published for the Polish Red breed, indicating an increasing reproductive rate through the maternal line. The average age at reproduction decreased to 1.64 years in bulls and 2.65 years in cows, suggesting a reduced selection and retention of bulls and an increased selection of cows, along with shorter herd tenure.

The selection of animals for reproduction drastically declined, with a 54.14% decrease in bulls and a 32.67% decrease in cows in the current population. The intensity of selection was 0.58% for males and 32.32% for females, reflecting a reduced interest among breeders in this breed or, for males, the utilization of males from other specialized crossbreeding breeds, similar to the Indubrasil breed [34].

The number of traced generations for both populations was seven, with Pedigree Completeness Index values greater than 0.5 in the first and second generations. While these values fall below the threshold of 0.6 for reliability in estimates, they have steadily improved since 2000, when the breed was included in the official catalog of Spanish bovine breeds, alongside enhanced management of genealogical records and genetic improvement plans by the association [2]. These values align with other bovine breeds undergoing conservation processes, such as the Curraleiro Pé Duro in Brazil, Italian Pontremolese, Sardo Modicana, and German Angler and Red and White dual-purpose breeds [36,37,38], but are lower than those in commercial–industrial dairy and beef cattle breeds with extensive record-keeping and genetic evaluation histories [39,40,41,42,43,44,45].

Figure 6 depicts the parameters GCI, F, ΔR, ΔF, and non-random mating, showing increases in all parameters from the year 2000. F, non-random mating, and GCI exhibit irregular increases, particularly in the periods 2003–2004, 2006–2007, and 2012–2013, due to the mating of closely related animals and increased genealogical information [26]. However, there has been a notable reduction in F since 2019 due to conservation policies [2]. The average value of F for the current population of the Andalusian Black cattle breed reaches 5.94%, which is significantly higher than other endangered breeds in Spain and Europe [46,47,48]. Higher values were reported in breeds like Mucca Pisana in Italy and Mertolenga in Portugal [27,37].

Variable GCI values have been observed across species, ranging from higher values of 9.11 to 9.65 for Spanish, Arabian, and Hispano–Arabian horse breeds [49] to similar values of 4.08 in the Brazilian Somali sheep breed [50]. These discrepancies can be attributed to GCI reliance on pedigree integrity and the number of generations, which are strongly influenced by conservation husbandry practices specific to each species, where genealogical registration has historically varied in importance.

For instance, the differing purposes and values of domestic animals play a significant role. Horses, often used for sport, leisure, or transportation, rely heavily on genealogical information for assessing abilities, traits, and value. Conversely, donkeys, presumed to have lost their original function, face challenges due to incomplete pedigrees. Poultry or rabbits, primarily used for meat production, prioritize factors like growth rate, feed efficiency, or carcass quality over genealogical information [51].

Additionally, variations exist in genetic diversity and domestication levels among domestic animals. Species like cattle, sheep, and goats, with a long history of domestication and selective breeding, benefit from genealogical information for tracing origins, relationships, and genetic improvement. In contrast, cats, dogs, and pigs, with a more recent and complex domestication history, may face challenges in obtaining or interpreting genealogical information if not registered in official studbooks [52].

Moreover, differences in legal and zootechnical frameworks for genealogical information registration across the European Union further contribute. Directives exist for the breeding and trade of equidae and purebred breeding animals [53], necessitating breed societies, studbooks, and identification systems. However, no such directives exist for hybrid breeding pigs or other domestic animals, leaving genealogical information registration at the discretion of Member States or breeders [53].

In the case of the Andalusian Black cattle breed, values have been less than 50% complete since the second generation. However, there has been an increase in GCI over time, indicating that conservation efforts towards registering genealogical information are becoming effective. This may facilitate decision-making for driving conservation strategies based on reliable parameters.

The ΔF, ΔR, and C metrics have exhibited a moderate increase from 1999, reaching a peak in 2016, followed by a trend toward reduction due to the implementation of genealogical record-keeping in 2012 (Figure 6). ΔF values were 2.71 and 3.17 for the historical and current populations, respectively, and they were comparable to values reported for other endangered Italian breeds like Sarda and Sarda Bruna [46]. However, compared to the Spanish Bruna dels Pirineus breed, ΔF was notably lower (0.99%) [46]. ΔR showed a value of 2.1%, similar to values found in the Mertolenga breed (2.05%) in Portugal and Colombian breeds like Horned Costeño, San Martinero, Blanco Orejinegro, and Romosinuano. Furthermore, ΔR values slightly exceeded those of C, which were 2.9% and 3.6% for the historical and current populations, respectively. Both C and ΔR values have displayed a downward trend since 2016, due to measures implemented to control parentage using genetic markers and limiting the use of breeding stock that endangers breed genetic diversity.

The non-random mating values (α) were 0.049 and 0.061 for the historical and current populations, respectively, with positive values observed since 2003, though experiencing a reduction since 2019. This suggests ineffective matings, with individuals having higher-than-average coancestry or relatedness mated more frequently, akin to observations in commercial breeds like Jersey in the USA. Additionally, α values were consistent with F, indicating a pattern of positive assortative mating when mating related individuals.

The effective number of equivalent founders (fe = 91.14) for the breed was comparable to that of the Mertolenga breed in Portugal (fe = 87.9), yet was much higher (+64%) than that reported for the Berrenda Negra breed in Spain (fe = 58) and lower than that of the Berrenda Colorada breed (fe = 140) in Spain. This suggests an imbalance in genetic contribution among founders, which is reflected in Table 4, where a small number of individuals account for a significant portion of the genetic diversity. Similarly, the effective number of ancestors (fa = 42) was lower than that of the Mertolenga breed and the Berrenda Negra and Colorada breeds. Both fe and fa are influenced by population size, indicating a genetically small population [27,54].

The fa/fe ratio in this study was 0.46, which was notably lower compared to several other breeds, including Berrenda Colorada, Berrenda Negra, Morucha, Mertolenga, Lageana, and Tabapua, which exhibited ratios greater than 1 [27,46,54,55,56,57]. Conversely, proportions in the Pinzgau cattle breed and the Marchigiana and Bosmara breeds in Brazil were similar, indicating a genetic bottleneck within the Negra Andaluza breed, affecting its genetic diversity, akin to observations in commercial dairy cattle breeds [58].

Historically, the Andalusian Black cattle breed may have faced a genetic bottleneck due to factors such as agricultural mechanization, crossbreeding, lack of genetic programs, subpopulation segmentation and isolation, and unstructured breeding efforts [4], contributing to population decline and genetic variation loss.

Equivalent founder genomes (fg) indicate the maintenance of founder genes in the population for a particular locus, revealing causes of gene loss through segregation [58]. In the Negra Andaluza breed, the fg value was 46.83, similar to the Mertolenga breed in Portugal [27]. The fg/fe ratio was 0.51, suggesting the impact of genetic drift, excluding founder contributions to genetic diversity, and was higher compared to the Simmental and Braunvieh breeds in Austria [59].

In terms of inbreeding and average relatedness increase in the reference population (both parents known), minimal values were observed (0.21% and 0.27%, respectively), indicating the positive effects of genealogical control measures and improvement plans, as evidenced by reductions in F and AR in recent years (Figure 6).

The Andalusian Black cattle breed exhibited a genetic diversity of 98.93% and a Genetic Conservation Index (total population) of 2.75, akin to other breeds under conservation programs such as Berrenda Negra and Berrenda Colorada in Spain [54]. However, the loss of genetic diversity was 1.07, which is relatively low compared to commercial dairy breeds experiencing losses between 3 and 11% [39].

Genetic diversity loss (GDL) due to genetic bottlenecks and drift was relatively low (1.07), being nearly double the GDL due to unequal founder contributions (0.55), as observed in the Guernsey breed [39]. In contrast, other conservation breeds like Berrenda Negra and Colorada exhibited lower GDL values [54], indicating genetic bottlenecks and drift as primary causes of genetic diversity loss in the Andalusian Black cattle breed.

Chakraborty [60] emphasizes the importance of understanding population ecology, demographics, and social behavior to detect population structure. This is particularly relevant for domestic and endangered populations, where anthropologic and administrative components must also be considered. F-statistics, describing inbreeding at the individual (F_IS_), subpopulation (F_ST_), and population (F_IT_) levels are commonly used for this purpose.

For the Andalusian Black breed, analysis was conducted at the herd (28), municipality (23), and provincial (5) levels. F_IS_ values were 0.036, 0.021, and 0.039, while F_ST_ was 0.085, 0.069, and 0.011 and F_IT_ was 0.052, 0.049, and 0.049, respectively. These values indicate a low-to-moderate level of inbreeding and differentiation at all levels, which is comparable to other breeds under conservation programs and intensive production breeds like Brown Swiss, Sahiwal, and Braford [36,61,62,63,64,65].

Positive but small F_IS_ values suggest some inbreeding within herds, municipalities, and provinces, while positive but small F_ST_ values indicate some differentiation among these levels. F_IT_ values, similar to F_IS_ values, suggest that within-population inbreeding contributes mainly to total inbreeding, with little effect from among-population differentiation [66]. These values reflect the positive effects of inbreeding control measures in the breed (Figure 6).

F_ST_ values close to 0 suggest similar allele frequencies at all levels, indicating a lack of genetic structure. A slight reduction in heterozygosity is observed at the provincial level, while greater reductions are seen at the municipality and herd levels. F_IT_ analysis reveals an average reduction of 4.9% in heterozygosity at the municipality and provincial levels and 5.2% at the herd level, which is comparable to findings in the Wagyu breed [67].

The inbreeding levels for herd, municipality, and province were 0.095, 0.078, and 0.022, respectively, similar to coancestry averages of 0.095, 0.079, and 0.022. These values indicate that subpopulations maintain low and adequate levels of inbreeding, aligning with recommendations for conservation.

The Nei distances show similarity at the herd and municipality levels (378 and 253, respectively) but considerable distance at the provincial level (10). Clade formation in Figure 7 and Figure 8 reveals distinct groupings: Cadiz and Jaen (0.047 and 0.016, respectively) and a third split into Sevilla and Cordoba–Huelva (0.010, 0.007, and 0.007, respectively). This suggests a limited genetic relationship between Cadiz and Jaen with Sevilla, Huelva, and Cordoba (which exhibit a greater genetic relationship).

At the municipality level, Barbate de Franco (Cadiz Province) is closely related to Espiel, Villanueva del Duque, and Cabra in Cordoba. Similarly, municipalities from Huelva and Sevilla appear alongside Cordoba, which are geographically close. In Jaen Province, only Baños de la Encina shows a relationship to municipalities in Sevilla and Huelva, which are also geographically proximate. 

The Western and Eastern Sorian Cañadas Reales, ancient routes used by the Mesta, an association of cow and sheep farmers practicing transhumance in Spain, span approximately 800 km and 700 km, respectively [68,69]. The Eastern route starts in Soria and extends to Sevilla, crossing territories inhabited by Andalusian Black cattle, such as Cádiz, Jaén, Sevilla, Córdoba, and Huelva. Although not directly connected, the Western Sorian intersects with the Cañada Real de la Plata near Guijuelo, Salamanca [69].

These routes form part of a network of ancient paths facilitating livestock movement between pastures throughout the seasons [70]. The Eastern Sorian Cañada Real links to provinces like Sevilla, Huelva, and Córdoba through vías pecuarias, which are reserved paths for livestock passage. These paths, including veredas and coladas, vary in width and length [55]. Examples include Vereda de Hornachuelos (20.89 m wide, 86.5 km long), Vereda de la Plata (20.89 m wide, 125.5 km long), and Vereda de la Sierra (20.89 m wide, 215 km long), and they connect different provinces through various towns like Hornachuelos, Palma del Río, Peñaflor, Lora del Río, Sanlúcar la Mayor, Villamanrique de la Condesa, and Aracena [70].

The Andalusian Black cattle belong to the Black Iberian group, alongside breeds like the Avileña–Negra Ibérica and the Preta [71]. Originating from the ancient Tronco Negro Ibérico, they share characteristics such as a black coat, medium size, and adaptability to harsh environments. Genetic studies suggest a closer relationship with Preta cattle than other Iberian breeds like the Black Avileña, possibly due to a shared genetic reservoir in the Aracena mountains, which are part of the Sierra Morena range along the Spain–Portugal border [71].

The Sierra Morena, connecting with Portugal’s Serra da Estrela, acts as a natural barrier between the countries and is home to the Aracena mountains, a protected area within the Sierra de Aracena and Picos de Aroche Natural Park. Despite high genetic diversity and low differentiation among these breeds, geographic isolation, environmental adaptation, and selective pressures may have contributed to some genetic differences, as reported for the same and other species [72,73,74].

Nei distances reveal moderate genetic differentiation among herds and municipalities of Andalusian Black cattle, which is likely influenced by geographic distance, limited gene flow, or local adaptation. However, there is minimal genetic variation among provinces, potentially due to historical transhumance or shared selection criteria.

When provinces are considered, genetic clustering illustrates relationships among them and their municipalities. Provinces like Sevilla, Huelva, and Córdoba show greater genetic relatedness, while Cádiz and Jaén exhibit less connection, which is possibly due to geographic isolation or environmental differences.

Territorial segregation plays a crucial role in preserving genetic diversity in endangered native cattle breeds. By restricting movement and interbreeding to specific regions, unique genetic traits can be maintained, preventing dilution of the gene pool from crossbreeding. This approach facilitates focused breeding programs tailored to local environments, ensuring the conservation of valuable genetic adaptations. Overall, territorial segregation supports the long-term survival and sustainability of endangered breeds. 

As part of future strategies, exploring the runs of homozygosity in the Andalusian Black cattle population using genomic data is recommended. This approach would significantly enhance the depth of the study and enable the validation of findings across different populations. Leveraging genomic data allows for a more comprehensive understanding of the genetic structure, diversity, and potential inbreeding patterns within this breed. Moreover, analyzing runs of homozygosity can provide valuable insights into the historical demography, selection pressures, and adaptation mechanisms of Andalusian Black cattle. Ultimately, integrating genomic analyses into the research framework will contribute to more informed conservation and breeding strategies aimed at preserving the unique genetic heritage of this endangered breed.

## 5. Conclusions

The Andalusian Black cattle breed has suffered a population decline due to factors like mechanization and foreign breed introduction. Initiatives such as the 2004 formation of the Association of Breeders of the Andalusian Black Cattle Breed and 2012 endangered status designation have boosted registrations. Nevertheless, the current population remains significantly reduced, and is influenced by foreign breeds and crossbreeding. Breeding practices indicate a decline in bulls and an increase in cows, potentially impacting genetic diversity. Selection trends favor cows over bulls, contributing to a high inbreeding coefficient (F). While the Genetic Conservation Index (GCI) has improved, it remains lower than for other breeds. Genetic diversity is imbalanced among founders, posing a risk of a bottleneck. Analysis at different levels reveals low-to-moderate inbreeding and differentiation. Genetic distances suggest some geographic influence, with provinces showing minimal differentiation. The municipalities located in the mountains of Aracena may have acted as a reservoir of genes of the populations where diversity was boosted around the end of Eastern and Western Sorian Cañadas Reales. Historical transhumance routes, like the Cañadas Reales, may have connected provinces. Geographic and ecological factors also influence genetic diversity. Challenges include historical bottlenecks, demographic changes, and potential reproductive impacts. Focus should be on maintaining genetic diversity, reducing inbreeding, and addressing demographic shifts. Continued genealogical registration, targeted breeding, and collaboration are crucial. Monitoring and adjusting selection practices for a balanced contribution from both sexes is vital for sustainable diversity.

## Figures and Tables

**Figure 1 animals-14-01131-f001:**
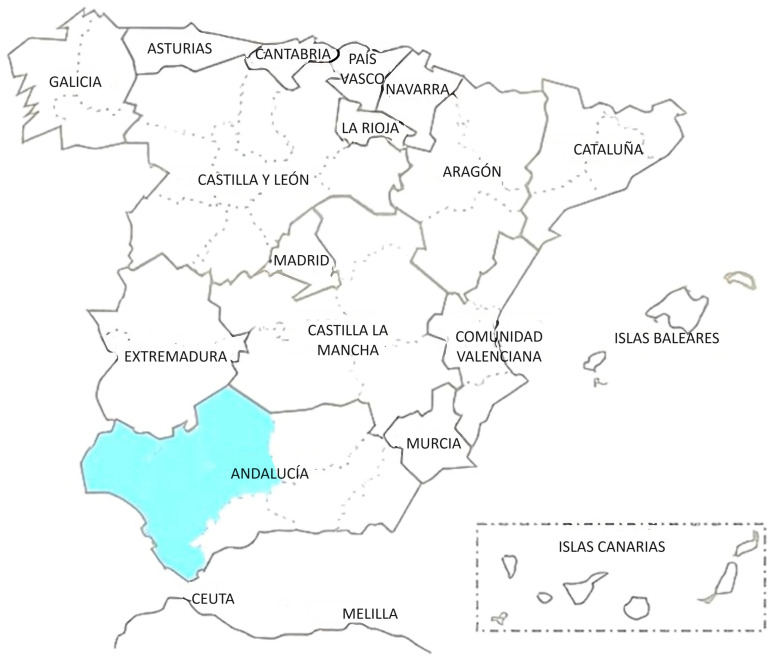
Area of expansion of Andalusian Black cattle breed (marked in turquoise).

**Figure 2 animals-14-01131-f002:**
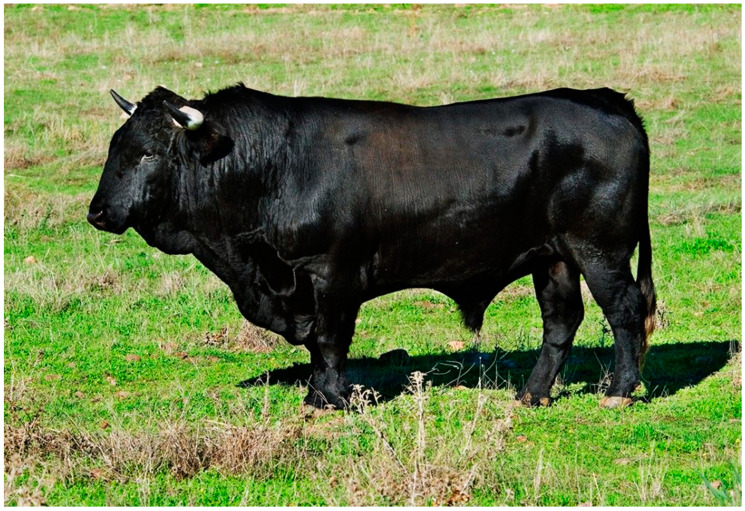
Andalusian Black bull. Picture by FEAGAS [5].

**Figure 3 animals-14-01131-f003:**
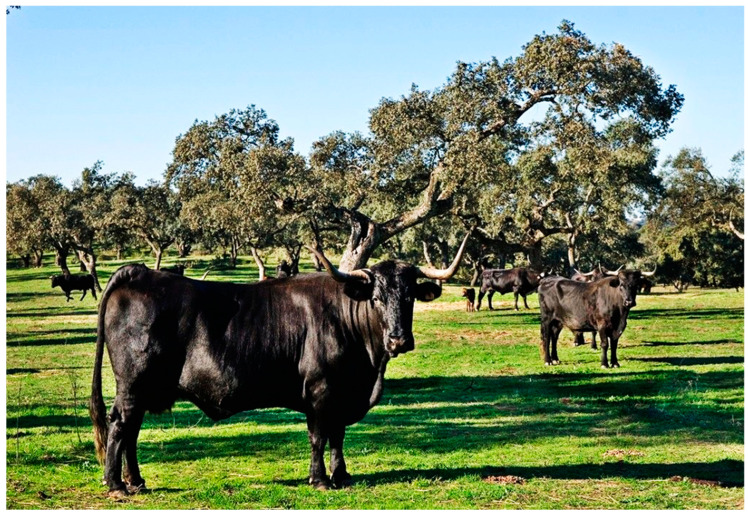
Andalusian Black cows. Picture by FEAGAS [5].

**Figure 4 animals-14-01131-f004:**
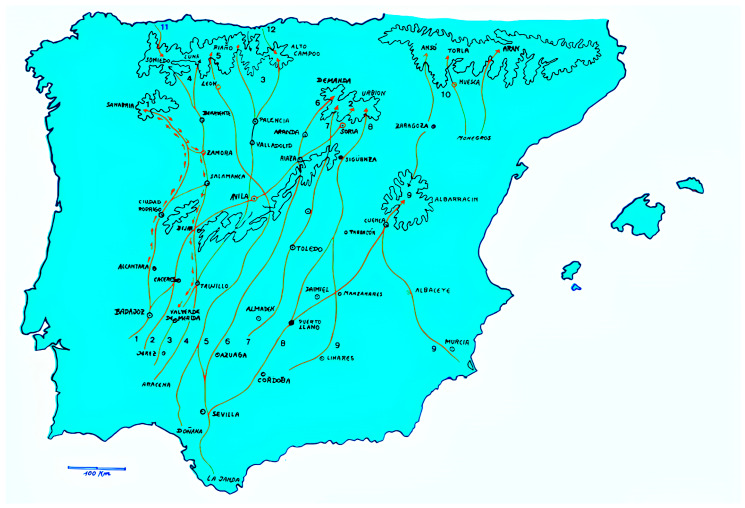
Cañadas Reales from the Iberian Peninsula. (1) Zamorana; (2) Soriana Occidental; (3) Leonesa Oriental; (4) De la Plata; (5) Leonesa Occidental; (6) Segoviana; (7) Galiana; (8) Soriana Oriental; (9) Conquense and Murciana; (10) Cabañeras of Aragon; (11) Vaqueiros de Alzada; and (12) Saja and Nansa. Red arrows are indicative of the routes followed by the animals.

**Figure 5 animals-14-01131-f005:**
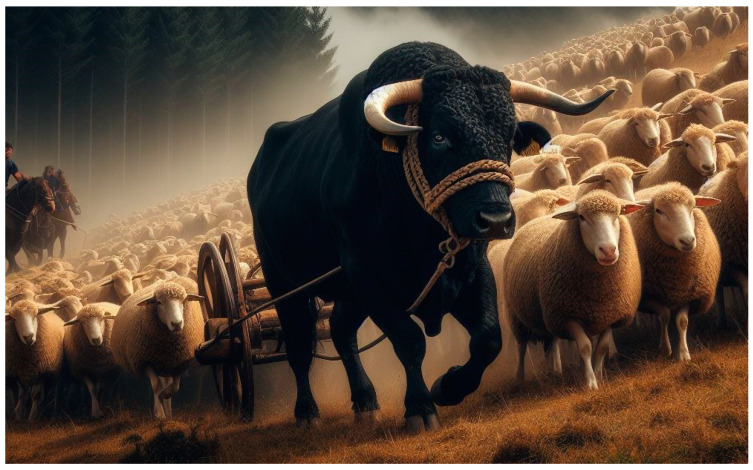
Andalusian Black cattle representation in its function as a draught animal.

**Figure 6 animals-14-01131-f006:**
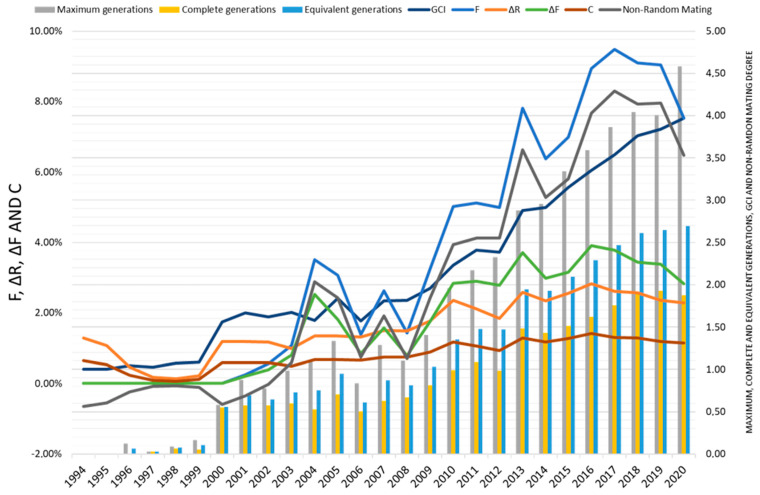
Evolution of the average coefficient of inbreeding (F, %), average coancestry (C, %), average individual increase in inbreeding (ΔF, %), average relatedness (∆R, %), and degree of non-random mating (α) of the Andalusian Black cattle breed population according to the number of maximum complete and equivalent generations from 1994 to 2020.

**Figure 7 animals-14-01131-f007:**
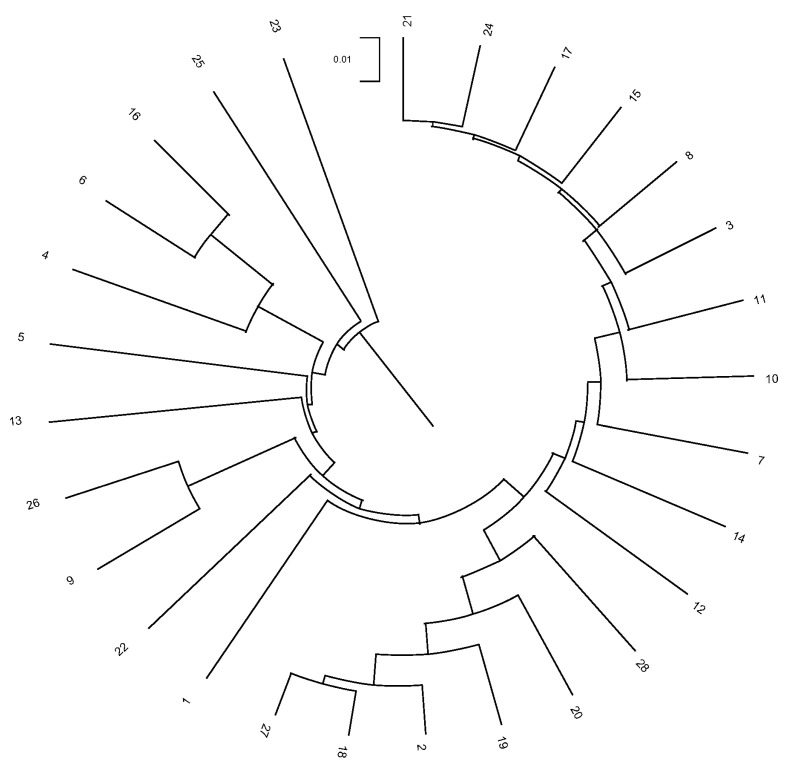
Dendrogram representing herd (encoded as numbers) relationships after computing Nei’s genetic distances.

**Figure 8 animals-14-01131-f008:**
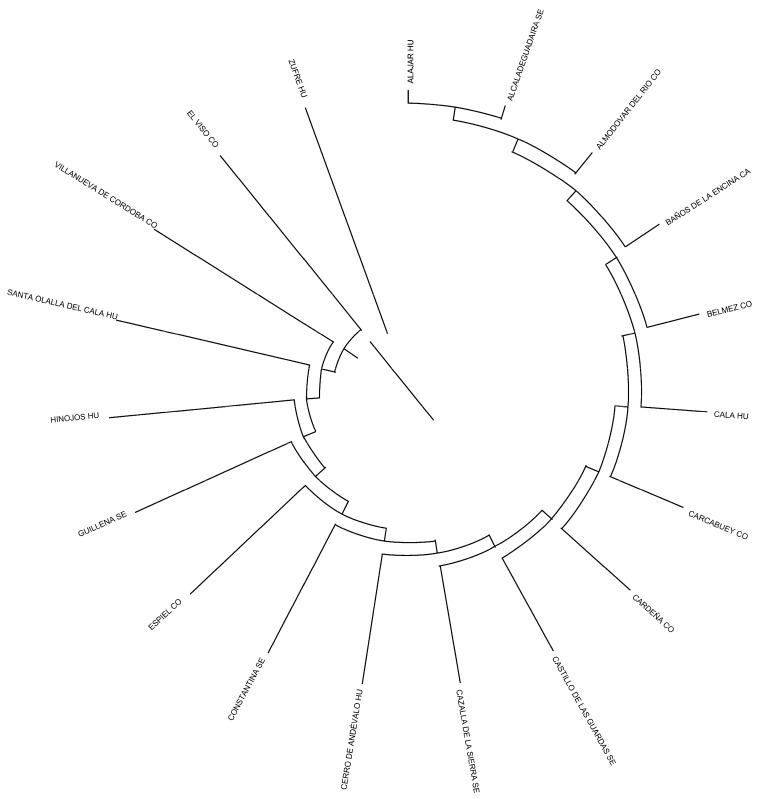
Dendrogram representing herds, municipalities, and provinces after computing Nei’s genetic distances. CO, Córdoba; HU, Huelva; SE, Sevilla; CA, Cádiz, respectively.

**Table 1 animals-14-01131-t001:** Summary of demographic and breeding-related statistics from historical (8555 dead and alive animals: 3593 bulls and 4962 cows—born between January 1994 and May 2020) and current (2472 alive animals: 593 bulls and 1.879 cows, born between February 2003 and May 2020) population pedigree datasets.

	Population Sets	Historical	Current
Parameters			
Total number of herds		28	28
Total number of provinces		5	5
Total number of municipalities		23	22
Average number of animals per herd/average herd size		305.54	88.28
Total bull percentage (%)		42.00%	23.99%
Mean number of calves per bull (n)		2.05	3.87
Maximum calf number per mated bull (n) (animals with unknown sire excluded)		379	282
Mean number of calves per mated bull (n) (animals with unknown sire excluded)		80.18	91.76
Average age of bull in reproduction (years)		9.51	7.87
Total cow percentage (%)		58.00	76.01
Mean number of calves per cow (n)		1.53	1.64
Maximum calf number per mated cow (n) (animals with unknown dam excluded)		14	13
Mean number of calves per mated cow (n) (animals with unknown dam excluded)		3.10	1.07
Average age of cows in reproduction (years)		11.24	8.59
Total cow/bull ratio		1.38/1	3.17/1
Mated cow/bull ratio		26.10/1	36.71/1
Progeny from bulls selected for breeding (%)		59.89	4.75
Progeny from cows selected for breeding (%)		59.39	26.67
Male selection intensity or portion of male calves born retained for breeding (%)		11.37	0.58
Female selection intensity or portion of female calves born retained for breeding (%)		40.57	32.32

**Table 2 animals-14-01131-t002:** Summary of statistics of population completeness level.

	Population Set	Historical	Current
Parameter			
Population size		8555	2472
Maximum number of traced generations (n)		7	7
Pedigree completeness level at 1st generation (Known parents)		0.8763	0.9846
Pedigree completeness level at 2nd generation (Known grandparents)		0.5538	0.7221
Pedigree completeness level at 3rd generation (Known great-grandparents)		0.2331	0.3260
Pedigree completeness level at 4th generation (Known great-great-grandparents)		0.0661	0.0871
Pedigree completeness level at 5th generation (Known great-great-great grandparents)		0.0098	0.0145
Number of maximum generations (mean ± SD)		2.65 ± 1.57	3.30 ± 1.32
Number of complete generations (mean ± SD)		1.27 ± 0.74	1.55 ± 0.63
Number of equivalent generations (mean ± SD)		1.74 ± 0.94	2.14 ± 0.73

**Table 3 animals-14-01131-t003:** Summary of pedigree analysis statistics.

	Populational Sets	Historical (n = 8555)	Current (n = 2472)
Parameter			
Inbreeding coefficient (F, %)		5.94	7.23
Average individual increase in inbreeding (ΔF, %)		2.71	3.17
Maximum coefficient of inbreeding (%)		43.75	43.75
Inbred animals (%)		37.80	48.82
Highly inbred animals (%)		0.193	0.185
Average coancestry coefficient (C, %)		2.90	3.60
Average relatedness coefficient (ΔR, %)		2.10	2.40
Non-random mating rate (α)		0.049	0.061
Genetic Conservation Index (GCI)		2.75	3.19

**Table 4 animals-14-01131-t004:** Probabilities of gene origin and founder analysis.

Parameter	Reference Population (Both Parents Known) (n = 2396)
Historical population	8555
Current population	2472
Base population (one or more unknown parents)	1204
Actual base population (one unknown parent = half founder)	1058
Number of founders contributing to the reference population (n)	985
Number of ancestors contributing to the reference population (n)	981
Effective number of non-founders (*Nef*)	96.30
Number of founder equivalents (*f_e_*)	91.14
Effective number of ancestors (*f_a_*)	42
Founder genome equivalents (*f_g_*)	46.83
fa/fe ratio	0.46
fg/fe ratio	0.51
Ancestors explaining 25% of the gene pool (n)	5
Ancestors explaining 50% of the gene pool (n)	17
Ancestors explaining 75% of the gene pool (n)	93
Average individual increase in inbreeding (ΔF, %)	0.21
Average relatedness (∆R, %)	0.27

**Table 5 animals-14-01131-t005:** Measures of genetic diversity and genetic diversity loss.

Parameter	Reference Population (Both Parents Known) (n = 2396)
Genetic diversity (GD, %)	98.93
Genetic diversity loss (GDL, %)	1.07
Genetic diversity in the reference population considered to compute the genetic diversity loss due to the unequal contribution of founders (DG*, %)	99.45
GDL due to bottlenecks and genetic drift since founders (GL, %)	1.07
GDL due to unequal founder contributions (%)	0.55

**Table 6 animals-14-01131-t006:** Wright’s fixation statistics and heterozygosity parameters when the subdivision criterion were herds, municipalities, and provinces.

Parameters	Herd/Breeder	Municipality	Province
Subpopulations ^a^	28	23	5
*F*_IS_ (Inbreeding coefficient relative to the subpopulation)	0.036	0.021	0.039
*F*_ST_ (Correlation between random gametes drawn from the subpopulation relative to the total population)	0.085	0.069	0.011
*F*_IT_ (Inbreeding coefficient relative to the total population)	0.052	0.049	0.049
Mean inbreeding within subpopulations	0.095	0.078	0.022
Mean number of animals per subpopulation	305.54	371.96	1711
Number of Nei genetic distances	378	253	10
Average Nei genetic distance	0.084	0.068	0.011
Mean coancestry within subpopulations	0.095	0.079	0.022
Self-coancestry	0.531	0.530	0.530
Mean coancestry in the metapopulation	0.012	0.011	0.011

^a^ Subpopulations/sublines can be determined if herds are clustered in different groups (that is, the number of herds and the number of populations disagree).

## Data Availability

Data will be made available from the corresponding author upon reasonable request.

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
