# Peer review of "Analysis of Endangered Andalusian Black Cattle (Negra Andaluza) Reveals Genetic Reservoir for Bovine Black Trunk"

_animals, 2024, doi:10.3390/ani14071131_

Round 1

Reviewer 1 Report

Comments and Suggestions for Authors

I appreciate the effort put into the study on the genetic analysis of endangered Andalusian Black Cattle. The findings hold significant potential for the scientific community. However, there are certain areas that should be addressed to improve the manuscript’s potential for publication.

1.          In this study, it appears that the same software package was used to assess inbreeding and population measures in Andalusian Black Cattle using pedigree data. However, for the interest of the readership, it’s essential to associate the inbreeding parameters with traits and/or breeding strategies to provide new insights.

2.          Is it relevant to include “Report to ………. Portugal” in the title of the manuscript?

3.          For Figure 1, use an appropriate text font for clarity and understanding for readers of all levels.

4.          Use high-resolution images for Figures 4 and 7 and ensure the text font is clear and readable.

5.          In section 2.1, the numbers and division of animals are confusing. Rewrite from Lines 170 to 175 for clarity.

6.          There is no information provided about the basics of pedigree information, such as the number of inbreds, founders, individuals with known parents, full-sib groups, and average family size. How did the authors check for pedigree errors? Please explain.

7.          The header of all tables, including supplementary files, is inappropriate format. Use the regular format of the table header as seen in other research articles.

8.          The author used pedigree information from 1994 to 2020, but Table 2 mentions only 5 generations. Please explain.

9.          Rewrite from lines 213 to 215 for clarity.

10.      Table 3 should be in the results section, not in the methods section.

11.      Why didn't the author check the inbreeding coefficient of the inbred animals?

12.      Provide a pedigree completeness index plot in the supplementary file.

13.      Remove the left part of the diagram from Figure 6 and place it in the supplementary file.

14.      I suggest shortening the Discussion section to 2 to 3 pages and include only relevant information related to this study.

Addressing these points will likely enhance the manuscript's quality and increase its chances of acceptance.

Author Response

I appreciate the effort put into the study on the genetic analysis of endangered Andalusian Black Cattle. The findings hold significant potential for the scientific community. However, there are certain areas that should be addressed to improve the manuscript’s potential for publication.

  1. In this study, it appears that the same software package was used to assess inbreeding and population measures in Andalusian Black Cattle using pedigree data. However, for the interest of the readership, it’s essential to associate the inbreeding parameters with traits and/or breeding strategies to provide new insights.

Response: We thank the reviewer for his/her kind comments and followed reviewer suggestions as follows..

  1. Is it relevant to include “Report to ………. Portugal” in the title of the manuscript?

Response: Yes, We think given it describes the basis of the paper.

  1. For Figure 1, use an appropriate text font for clarity and understanding for readers of all levels.

Response: We followed reviewer suggestion.

  1. Use high-resolution images for Figures 4 and 7 and ensure the text font is clear and readable.

Response: We followed reviewer suggestion.

  1. In section 2.1, the numbers and division of animals are confusing. Rewrite from Lines 170 to 175 for clarity.

Response: We followed reviewer suggestion.

  1. There is no information provided about the basics of pedigree information, such as the number of inbreds, founders, individuals with known parents, full-sib groups, and average family size. How did the authors check for pedigree errors? Please explain.

Response: This information is provided across the Tables. For example, number of inbreds (Table 3), founders (Table 4), individuals with known parents (Table 4), full-sib groups (in body text) and average family size (Table 6). Furthermore, pedigree errors were checked and revised by ENDOG software as if the pedigree file is not coherent, the software does not run. Then this were corrected manually as each case may need for a specific manner to correct. In these regards the most common errors found are sex misattributions, filiation errors, age mistakes and animal naming. We clarified this in the body text.

  1. The header of all tables, including supplementary files, is inappropriate format. Use the regular format of the table header as seen in other research articles.

Response: We checked it and there are many articles in which this format has been used. Eiher in other papers from MDPI or other editorials. We understand the reviewer concern, anyway if there is a need to change it, editor will request us.

  1. The author used pedigree information from 1994 to 2020, but Table 2 mentions only 5 generations. Please explain.

Response: Table 2 reports the number of maximum, equivalent and complete generations. Then reports pedigree completeness level up to 5th generation which is the one from which completeness levels go around 0% on average. Normally in animals no further than 5th generation is provided as shown in literature, due to the aforementioned reasons.

  1. Rewrite from lines 213 to 215 for clarity.

Response: We followed reviewer suggestion.

  1. Table 3 should be in the results section, not in the methods section.

Response: We followed reviewer suggestion.

  1. Why didn't the author check the inbreeding coefficient of the inbred animals?

Response: Sorry, this information is provided in Table 3.

  1. Provide a pedigree completeness index plot in the supplementary file.

Response: We followed reviewer suggestion.

  1. Remove the left part of the diagram from Figure 6 and place it in the supplementary file.

Response: We followed reviewer suggestion.

  1. I suggest shortening the Discussion section to 2 to 3 pages and include only relevant information related to this study.

 Response: Discussion has been shortened from 4764 words to 2775.

Addressing these points will likely enhance the manuscript's quality and increase its chances of acceptance.

Response: Again, we thank the reviewer for his/her kind comments, time and attention to our manuscript.

Reviewer 2 Report

Comments and Suggestions for Authors

The study provides valuable data on the genetic analysis of Endangered Andalusian Black Cattle. These findings can be particularly useful in addressing the demographic shifts and preserving genetic diversity in the future. I have a few minor comments that need to be included, as follows:

1. Line 395, In Figure 7, What do these numbers represent? Is it a cattle farm?

2. For the Tables, I suggest the author explains "current" what year it represents, as it is misleading.

3. Finally, I would suggest to explore the runs of homozygosity in this population using genomic data in future studies. This would add value in the study and make it feasible to test the findings in other populations as well.

Comments on the Quality of English Language

The study provides valuable data on the genetic analysis of Endangered Andalusian Black Cattle. These findings can be particularly useful in addressing the demographic shifts and preserving genetic diversity in the future. I have a few minor comments that need to be included, as follows:

1. Line 395, In Figure 7, What do these numbers represent? Is it a cattle farm?

2. For the Tables, I suggest the author explains "current" what year it represents, as it is misleading.

3. Finally, I would suggest to explore the runs of homozygosity in this population using genomic data in future studies. This would add value in the study and make it feasible to test the findings in other populations as well.

Author Response

Reviewer 2

The study provides valuable data on the genetic analysis of Endangered Andalusian Black Cattle. These findings can be particularly useful in addressing the demographic shifts and preserving genetic diversity in the future. I have a few minor comments that need to be included, as follows:

Response: We thank the reviewer for his/her kind comments, time and attention to our manuscript.

  1. Line 395, In Figure 7, What do these numbers represent? Is it a cattle farm?

Response: Yes, we clarified it.

  1. For the Tables, I suggest the author explains "current" what year it represents, as it is misleading.

Response: Clarified.

  1. Finally, I would suggest to explore the runs of homozygosity in this population using genomic data in future studies. This would add value in the study and make it feasible to test the findings in other populations as well.

Response: We added this at the end of discussion as a future strategy to perform.

Reviewer 3 Report

Comments and Suggestions for Authors

This manuscript is about the analysis of the endangered Andalusian Black Cattle. The study sheds light on the genetic reservoir for bovine black trunks in Northern Spain and Portugal, highlighting the challenges faced by this unique breed. The research provides answers to different and important questions such as, what are the main factors contributing to the decline in population of the Black Andalusian cattle breed? How has the introduction of foreign breeds impacted the average herd size of the Andalusian Black Cattle? What are the implications of this study for conservation efforts and the preservation of genetic diversity in livestock?

The manuscript is very well written, has the potential to be widely cited for bringing a problem that covers most of the indigenous races, and records part of the history of this important breed.

I suggest adjusting the Title reducing this only to the First part: Analysis of Threatened Andalusian Black Cattle Reveals Genetic Reservoir for Black Bovine Trunk.

Simple Summary is Larger than the Abstract, something that is usually the other way around. I suggest reducing the Simple Summary.

Figure 7 needs to be improved for dendrogram numbering size.

Author Response

This manuscript is about the analysis of the endangered Andalusian Black Cattle. The study sheds light on the genetic reservoir for bovine black trunks in Northern Spain and Portugal, highlighting the challenges faced by this unique breed. The research provides answers to different and important questions such as, what are the main factors contributing to the decline in population of the Black Andalusian cattle breed? How has the introduction of foreign breeds impacted the average herd size of the Andalusian Black Cattle? What are the implications of this study for conservation efforts and the preservation of genetic diversity in livestock?

The manuscript is very well written, has the potential to be widely cited for bringing a problem that covers most of the indigenous races, and records part of the history of this important breed.

Response: We thank the reviewer for his/her kind comments, time and attention to our manuscript.

I suggest adjusting the Title reducing this only to the First part: Analysis of Threatened Andalusian Black Cattle Reveals Genetic Reservoir for Black Bovine Trunk.

Response: We followed reviewer suggestion.

Simple Summary is Larger than the Abstract, something that is usually the other way around. I suggest reducing the Simple Summary.

Response: We followed reviewer suggestion.

Figure 7 needs to be improved for dendrogram numbering size.

Response: We followed reviewer suggestion.

Reviewer 4 Report

Comments and Suggestions for Authors

This manuscript reports a detailed study of the structure of an indigenous Spanish cattle population based on demographic data and pedigree information. The topic is interesting and of great practical importance for the management of a limited diffusion population.

Indeed, a few, but important concerns must be addressed before it can be considered for publication:

1) The size of the population studied is unclear. In the introduction section, it is reported "a population of 1,797 animals in 22 farms", but on line 147 it is reported "all alive animals .......  2,472 animals, 593 bulls and 1,879 cows".

Furthermore, in Table 1 there are 28 farms.

The number that can be deduced from the registers of the Spanish Ministry of Agriculture (Datos Censales NEGRA ANDALUZA) and from the FAO database (DAD-IS) is 2210 animals for the year 2020. Please clarify the issue precisely.

2) The manuscript is excessively long, especially in part of the introduction and discussion section. For example, lines 403 - 416 are a repetition of elements present in other parts of the manuscript.

L 681 - 732: the concept can be expressed in a few lines.

3) Throughout the text: always adopt the same indication for bibliographic references i.e. [n] and not (author, year).

minor issues:

The breed name in the original language could be added, perhaps in the title [Andalusian Black Cattle (Negra Andaluza)].

L 74: figure 1 shows the diffusion of the breed.

L 110 - 117: this part is too similar to what is reported in "https://academia-lab.com/encyclopedia/council-of-the-mesta/", if you quote literally, insert the text in quotation marks and report the source. Otherwise, rewrite completely. Check all text.

L 132: figure 5 does not seem very suitable to represent the expressed concept. It seems more appropriate to what is written in L121 – 123 or L135 - 136.

L 182: normally only the first author is reported (also the last but only if there are only two authors). Check all text.

Table 1: insert the measurement units between brackets, (%), (n), ... and so on.

Figure 6: these are the substantial results of the study, they should be presented with maximum definition and readability. Separate the two graphs and enlarge the one showing the time trends of the different calculated parameters. The succession of breed management and protection actions can be presented horizontally.

Figures 7 and 8: the dendrograms are illegible, I recommend increase the label size and use different colors to indicate, for example, which municipality the different farms belong to.

L 554 - 556, L 577 - 580: insert bibliographical references and check the entire text.

L 733 - 734: This is not demonstrated in this study.

L 758: This statement is very questionable. Please explain better or delete it.

Author Response

This manuscript reports a detailed study of the structure of an indigenous Spanish cattle population based on demographic data and pedigree information. The topic is interesting and of great practical importance for the management of a limited diffusion population.

Response: We thank the reviewer for his/her kind comments, time and attention to our manuscript.

Indeed, a few, but important concerns must be addressed before it can be considered for publication:

  • The size of the population studied is unclear. In the introduction section, it is reported "a population of 1,797 animals in 22 farms", but on line 147 it is reported "all alive animals .......  2,472 animals, 593 bulls and 1,879 cows".

Response: We clarified this in the body text.

Furthermore, in Table 1 there are 28 farms.

Response: We clarified this in the body text. One thing is the number that it is officially declared, and the other the number of herds that participated in the study. For example owners that are officially and administratively declared as one but on field opérate as two different herds.

The number that can be deduced from the registers of the Spanish Ministry of Agriculture (Datos Censales NEGRA ANDALUZA) and from the FAO database (DAD-IS) is 2210 animals for the year 2020. Please clarify the issue precisely.

Response: Same applies. Sometimes animals are registered in the studbook of the breed but they are not administratively declared due to the specific requeirements for each breed. For example, sometimes young animals may not be adminsitratively considered.

  • The manuscript is excessively long, especially in part of the introduction and discussion section. For example, lines 403 - 416 are a repetition of elements present in other parts of the manuscript.

Response: Discussion has been shortened from 4764 words to 2775.

L 681 - 732: the concept can be expressed in a few lines.

Response: We followed reviewer suggestion.

3) Throughout the text: always adopt the same indication for bibliographic references i.e. [n] and not (author, year).

 Response: We followed the reviewer suggestion.

minor issues:

The breed name in the original language could be added, perhaps in the title [Andalusian Black Cattle (Negra Andaluza)].

 Response: We followed the reviewer suggestion.

L 74: figure 1 shows the diffusion of the breed.

 Response: We followed the reviewer suggestion.

L 110 - 117: this part is too similar to what is reported in "https://academia-lab.com/encyclopedia/council-of-the-mesta/", if you quote literally, insert the text in quotation marks and report the source. Otherwise, rewrite completely. Check all text.

Response: We rewrote the text as suggested by the reviewer.

L 132: figure 5 does not seem very suitable to represent the expressed concept. It seems more appropriate to what is written in L121 – 123 or L135 - 136.

 Response: We followed the reviewer suggestion and relocated it.

L 182: normally only the first author is reported (also the last but only if there are only two authors). Check all text.

 Response: We followed the reviewer suggestion.

Table 1: insert the measurement units between brackets, (%), (n), ... and so on.

Response: We followed the reviewer suggestion.

Figure 6: these are the substantial results of the study, they should be presented with maximum definition and readability. Separate the two graphs and enlarge the one showing the time trends of the different calculated parameters. The succession of breed management and protection actions can be presented horizontally.

Response: We followed the reviewer suggestion. Graph in the left was moved to Supplementary Material as suggested by other reviewer.

Figures 7 and 8: the dendrograms are illegible, I recommend increase the label size and use different colors to indicate, for example, which municipality the different farms belong to.

Response: We followed the reviewer suggestion.

L 554 - 556, L 577 - 580: insert bibliographical references and check the entire text.

Response: We followed the reviewer suggestion.

L 733 - 734: This is not demonstrated in this study.

Response: We followed the reviewer suggestion and removed it.

L 758: This statement is very questionable. Please explain better or delete it.

Response: We followed the reviewer suggestion and removed it.

Round 2

Reviewer 1 Report

Comments and Suggestions for Authors

Dear Authors, I have reviewed your manuscript and commend the revisions made according to the comments and suggestions provided during the first revision. I appreciate your thorough attention to detail in addressing the comments, which significantly improved the quality and clarity of the manuscript. Based on the revisions, I am pleased to accept the article for publication. Thank you for your efforts.

Author Response

Dear Authors, I have reviewed your manuscript and commend the revisions made according to the comments and suggestions provided during the first revision. I appreciate your thorough attention to detail in addressing the comments, which significantly improved the quality and clarity of the manuscript. Based on the revisions, I am pleased to accept the article for publication. Thank you for your efforts.

Response: We thank the reviewer for his/her kind comments.

Reviewer 4 Report

Comments and Suggestions for Authors

Almost all of my concerns have been addressed and the manuscript can be accepted in its current form with only a few minor changes.

My comment on line 74 (now 66) was misunderstood. The “Area of Expansion” expression was fine, what wasn't fine was the location of the reference in the text.

In line 66 the morphological traits are illustrated, the reference figures are therefore 2 and 3.

While lines 46 - 49 illustrate the current diffusion of the breed. In this case figure 1 is appropriate.

Figure 5 also keeps being introduced in the text in the wrong place!

L 164: remove the minus sign in front of the number of bulls (also in line 196).

L 205: figure S2 must be introduced after S1. Rename the supplementary files and fix the text.

Author Response

Almost all of my concerns have been addressed and the manuscript can be accepted in its current form with only a few minor changes.

Response: We thank the reviewer for his/her kind comments.

My comment on line 74 (now 66) was misunderstood. The “Area of Expansion” expression was fine, what wasn't fine was the location of the reference in the text.

Response: we apologize for the misunderstanding and changing back to Area of Expansion.

In line 66 the morphological traits are illustrated, the reference figures are therefore 2 and 3.

Response: We changed it accordingly.

While lines 46 - 49 illustrate the current diffusion of the breed. In this case figure 1 is appropriate.

Response: We changed it accordingly.

Figure 5 also keeps being introduced in the text in the wrong place!

Response: We changed it accordingly.

L 164: remove the minus sign in front of the number of bulls (also in line 196).

Response: Removed.

L 205: figure S2 must be introduced after S1. Rename the supplementary files and fix the text.

Response: Reviewer suggestion was followed.